# Safety and immunogenicity following a homologous booster dose of CoronaVac in children and adolescents

Lei Wang [1,2,8], Zhiwei Wu[3,8], Zhifang Ying[4,8], Minjie Li [3], Yuansheng Hu[2], Qun Shu[5], Jing Li[6], Huixian Wang[7], Hengming Zhang[2], Wenbin Jiao[7], Lin Wang [6], Yuliang Zhao [3,9] ✉ & Qiang Gao [6,9] ✉

Data on safety and immunity elicited by a third booster dose of inactivated COVID-19 vaccine in children and adolescents are scarce. Here we conducted a study based on a double-blind, randomised, placebo-controlled phase 2 clinical trial (NCT04551547) to assess the safety and immunogenicity of a third dose of CoronaVac. In this study, 384 participants in the vaccine group were assigned to two cohorts. One received the third dose at a 10-months interval (cohort 1) and the other one at a 12-months interval (cohort 2). The primary endpoint is safety and immunogenicity following a third dose of CoronaVac. The secondary endpoint is antibody persistence following the primary two-dose schedule. Severities of local and systemic adverse reactions reported within 28 days after dose 3 were mild and moderate in both cohorts. A third dose of CoronaVac increased GMTs to 681.0 (95%CI: 545.2–850.7) in cohort 1 and 745.2 (95%CI: 577.0–962.3) in cohort 2. Seropositivity rates against the prototype were 100% on day 28 after dose 3. Seropositivity rates against the Omicron variant were 90.6% (cohort 1) and 91.5% (cohort 2). A homologous booster dose of CoronaVac is safe and induces a significant neutralising antibody levels increase in children and adolescents.

On November 24, 2021, the novel SARS-CoV-2 Omicron (B.1.1.529) variant was first reported to WHO from South Africa[1], and it has rapidly replaced the highly transmissible Delta (B.1.617.2) variant as the predominant variant worldwide[2]. The significant immune escape of the Omicron variant in convalescent patients infected with the prototype SARS-CoV-2 and other variants of concern (VOC) was reported[3–5]. The levels of neutralising antibodies against the Omicron are low and only short-lived after two-dose primary vaccination, while are enhanced with a third dose booster dose in adults[6–10].

Studies also found that three doses of inactivated COVID-19 vaccine could induce the cross-neutralising potency against the Omicron

variant in adults[11,12]. In our previous study, two doses of inactivated COVID-19 vaccine (CoronaVac, Sinovac Life Sciences Co., Ltd) induced higher neutralising antibody concentrations in children and adolescents aged 3–17 years compared with the adults[13–15]. At present, individuals 12 years of age and older are eligible for a single booster dose of the Pfizer-BioNTech COVID-19 Vaccine, Bivalent if it has been at least two months since they have completed primary vaccination[16]. However, Data on immunity and safety elicited by a third booster dose of inactivated COVID-19 vaccine in children and adolescents are scarce. Few countries approve of booster doses in this population. In our previous study, antibody levels in children were higher than those in

[1]National Immunisation Programme, Chinese Center for Disease Control and Prevention, Beijing, China. [2]Sinovac Biotech Ltd., Beijing, China. [3]Hebei Provincial Center for Disease Control and Prevention, Shijiazhuang, Hebei Province, China. [4]National Institutes for Food and Drug Control, Beijing, China. [5]Beijing Key Tech Statistics Technology Co., Ltd., Beijing, China. [6]Sinovac Life Sciences Co., Ltd., Beijing, China. [7]Zanhuang County Center for Disease Control and Prevention, Zanhuang, Hebei Province, China. [8]These authors contributed equally: Lei Wang, Zhiwei Wu, Zhifang Ying. [9]These authors jointly supervised this work: Yuliang Zhao, Qiang Gao. ✉e-mail: yuliang_zh1@163.com; gaoq@sinovac.com

adults and the elderly at 6–8 months after a primary schedule[17]. Due to the waning antibody levels over time, whether giving a booster with 10–12 months interval is reasonable in children and adolescents should be evaluated. It is unclear whether the immunity and safety elicited by a booster dose of inactivated COVID-19 vaccine are maintained for the Omicron variant in this population.

To fill this knowledge gap, we assessed the immune persistence after primary immunisation with CoronaVac, and the safety, immunogenicity, especially the cross-neutralising activity against the Omicron variant after a third dose of CoronaVac in children and adolescents aged 3–17 years.

## Results

Between December 12 and December 20, 2020, 515 individuals were screened, 480 participants were enroled in phase 2 trial, of whom 96 were randomly allocated to the placebo group, and 384 were randomly allocated to the 1.5 µg or 3.0 µg vaccine group. All 96 participants in placebo groups withdrew from the study and received COVID-19 vaccines since the inactivated COVID-19 vaccine was approved for emergency use among children and adolescents aged 3–17 years[18]. Of these 384 participants, 192 (50%) participants were allocated to cohort 1, and 192 (50%) participants were allocated to cohort 2. In cohort 1, 180 (94%) participants completed blood sampling at 3 months, and 171 (89%) participants completed blood sampling at 10 months after the second dose. In cohort 2, 185 (96%) participants completed blood sampling at 6 months, and 175 (91%) participants completed blood sampling at 12 months after the second dose. 171 participants in cohort 1 (89% of the 192 participants eligible for a third dose) received a third dose at a median of 300 days (range: 300–329 days) after the second dose. 175 participants in cohort 2 (91% of the 192 participants eligible for a third dose) received a third dose at a median of 363 days (range: 361–390 days) after the second dose. 170 (89%) participants in cohort 1 and 169 (88%) participants in cohort 2 completed blood sampling 28 days after the third dose for immunogenic evaluation (Fig. 1).

Between 13 November 2021 and 11 February 2022, All 346 (90%) participants who received the third dose were included in the safety analyses, and 337 (88%) were included in the per-protocol population for the immunogenic evaluation on day 28 after the third dose. Nine participants were excluded because two received other licensed inactivated COVID-19 vaccine BBIBP-CorV (Beijing Institute of Biological Products, Beijing, China). Seven did not have blood samples taken on day 28 (with a 10-day window period) after the third dose (Fig. 1). Mean age of participants were between 9.0 years (SD 3.7) to 9.4 years (SD 3.7) in cohort 1 and 9.3 years (SD 3.8) to 9.4 years (SD 3.8) in cohort 2 (Table 1). At baseline, none of the participants in any cohort had detectable neutralising antibodies (Fig. 2). From the primary vaccinations, no natural infections were reported in any cohort.

The immune persistence analysis from cohort 1 showed that, by 3 months after the second dose, the geometric mean titre (GMT) of neutralising antibody against prototype SARS-CoV-2 in the 1.5 µg group was 67.4 (95% CI 55.0–82.5) and seropositivity remained above 98.8% (95% CI 93.7–99.97%). GMT in the 3.0 µg group was 110.3 (95% CI 90.4–134.6) and seropositivity remained 100% (95% CI 95.8–100.0%). By 10 months after the second dose, the GMT dropped to 12.7 (95% CI 10.4–15.5) and 20.8 (95% CI 16.5–26.1) in the 1.5 µg group and 3.0 µg group, respectively (Fig. 2a; Supplementary Table 3-1). A third dose of CoronaVac given at month 10 after the second dose significantly increased neutralising antibody levels by the first two doses. In the 1.5 µg group, the GMT on day 28 after the second dose was 81.8 (95% CI 64.3–104.1), and on day 28 after the third dose was 597.7 (95% CI 452.4–789.6). While in the 3.0 µg group, the GMT on day 28 after the second dose was 138.1 (95% CI 111.9–170.3), and on day 28 after the third dose was 681.0 (95% CI 545.2–850.7). Neutralising antibody concentrations on day 28 after booster (Day 356) were approximately 7.3-fold (95% CI 5.3–10.0) higher than neutralising antibody

concentrations on day 28 after the second dose (Day 56) in the 1.5 µg group and approximately 4.9-fold (95% CI 3.7–6.4) in the 3.0 µg group (Supplementary Table 3-1). Geometric mean fold rise (GMFR) of neutralising antibodies from pre-booster to 28 days after the third dose were 46.4 (95% CI 35.3–60.9) for the 1.5 µg group and 32.8 (95% CI 25.1–42.8) for the 3.0 µg group. Seropositivity rates in both vaccination groups were 100% on day 28 after the booster dose (Fig. 2a; Table 2).

In the immune persistence analysis of cohort 2, in the 1.5 µg group, neutralising antibody titers against prototype SARS-CoV-2 declined to 22.1 (95% CI 18.1–26.9) at 6 months and 16.4 (95% CI 13.0–20.6) at 12 months after the second dose. In 3.0 µg group, GMT declined to 27.2 (95% CI 22.7–32.5) at 6 months and 21.7 (95% CI 18.1–26.0) at 12 months after dose 2. Seropositivity rate remained 76.7% (95% CI 66.4–85.2) in 1.5 µg and 93.1% (95% CI 85.6–97.4) in 3.0 µg group at 12 months (Fig. 2b; Supplementary Table 3-1). After administering a third dose of CoronaVac at 12 months, GMT increased to 462.4 (95% CI 364.4–586.6) in the 1.5 µg group and 745.2 (95% CI 577.0–962.3) in the 3.0 µg group on day 28 after the booster dose, and there was a significant difference between the two dose groups (p = 0.007). Neutralising antibody titers on day 28 after dose 3 (Day 416) were approximately five-fold higher than netralising antibody titers on day 28 after dose 2 (Day 56) in the 3.0 µg (from a GMT of 147.4–745.2, Fig. 2b) in cohort 2. The GMFR of neutralising antibodies from before to after the booster dose were 28.0 (95% CI 22.0–35.6) in the 1.5 µg group and 33.3 (95% CI 25.0–44.4) in the 3.0 µg group. The seropositivity rate in both vaccination groups in cohort 2 were 100% on day 28 after dose 3 (Fig. 2b; Table 2).

In the analysis of cross-neutralising activity against the Omicron variant, after administering a booster at 10 months after dose 2 in cohort 1, GMT increased to 23.6 (95% CI 17.7–31.4) in the 1.5 µg group and 34.5 (95% CI 27.2–43.7) in 3.0 µg group 28 days later. The GMFR from before to after the booster dose were 11.5 (95% CI 8.6–15.2) in the 1.5 µg group and 16.0 (95% CI 12.8–20.0) in the 3.0 µg group (Fig. 3a, Table 2). Similarly, in cohort 2, after administering a booster at 12 months, GMT increased to 24.0 (95% CI 18.0–32.2) in the 1.5 µg group and 40.0 (95% CI 30.7–52.3) in the 3.0 µg group 28 days later. The GMFR from before to after the booster dose were 11.2 (95% CI 8.4–14.8) in the 1.5 µg group and 19.2 (95% CI 14.8–24.8) in the 3.0 µg group (Fig. 3b, Table 2).

Further analysis of the GMTs against prototype and Omicron strain showed that the GMT of neutralising antibody titers against the Omicron variant was 23.6 (95%CI 17.7–31.4), which 25.3-fold decreased compared with the prototype strain (GMT 597.7, [95%CI 452.4–789.6]) in the 1.5 µg group in cohort 1. However, GMTs of the 3.0 µg group were around 19.7-fold lower against the Omicron variant (GMT 34.5, [95% CI 27.2–43.7]) than against the prototype strain (GMT 681.0, [95% CI 545.2–850.7]) (Fig. 4). Seropositivity rate against the Omicron was 77.7% (95% CI 67.3–86.0) in the 1.5 µg group and 90.6% (95% CI 82.3–95.9) in 3.0 µg (Table 2). In cohort 2, GMTs of 1.5 µg group were around 19.2-fold lower against the Omicron variant (GMT 24.0, [95% CI 18.0–32.2]) than against prototype strain (GMT 462.4, [95% CI 364.4–586.6]). GMTs of 3.0 µg group were around 18.6-fold lower against the Omicron variant (GMT 40.0, [95% CI 30.7–52.3]) than against the prototype strain (GMT 745.2, [95% CI 577.0–962.3]). The seropositivity rate against the Omicron was 78.8% (95% CI 68.6–86.9) in 1.5 µg group and 91.5% (95% CI 83.2–96.5) in 3.0 µg (Table 2).

In an exploratory analysis by age, seropositivity rates against the Omicron variant on day 28 after the third dose of 1.5 µg or 3.0 µg in both cohorts were 100% in participants aged 3–5 years. The GMTs of participants aged 3–5 years were higher than the 6–11 years group and the 12–17 years group in any doses. There was a significant difference among the three age groups and a negative correlation between age and neutralising antibody titers against the Omicron variant (Supplementary Table 3-2).

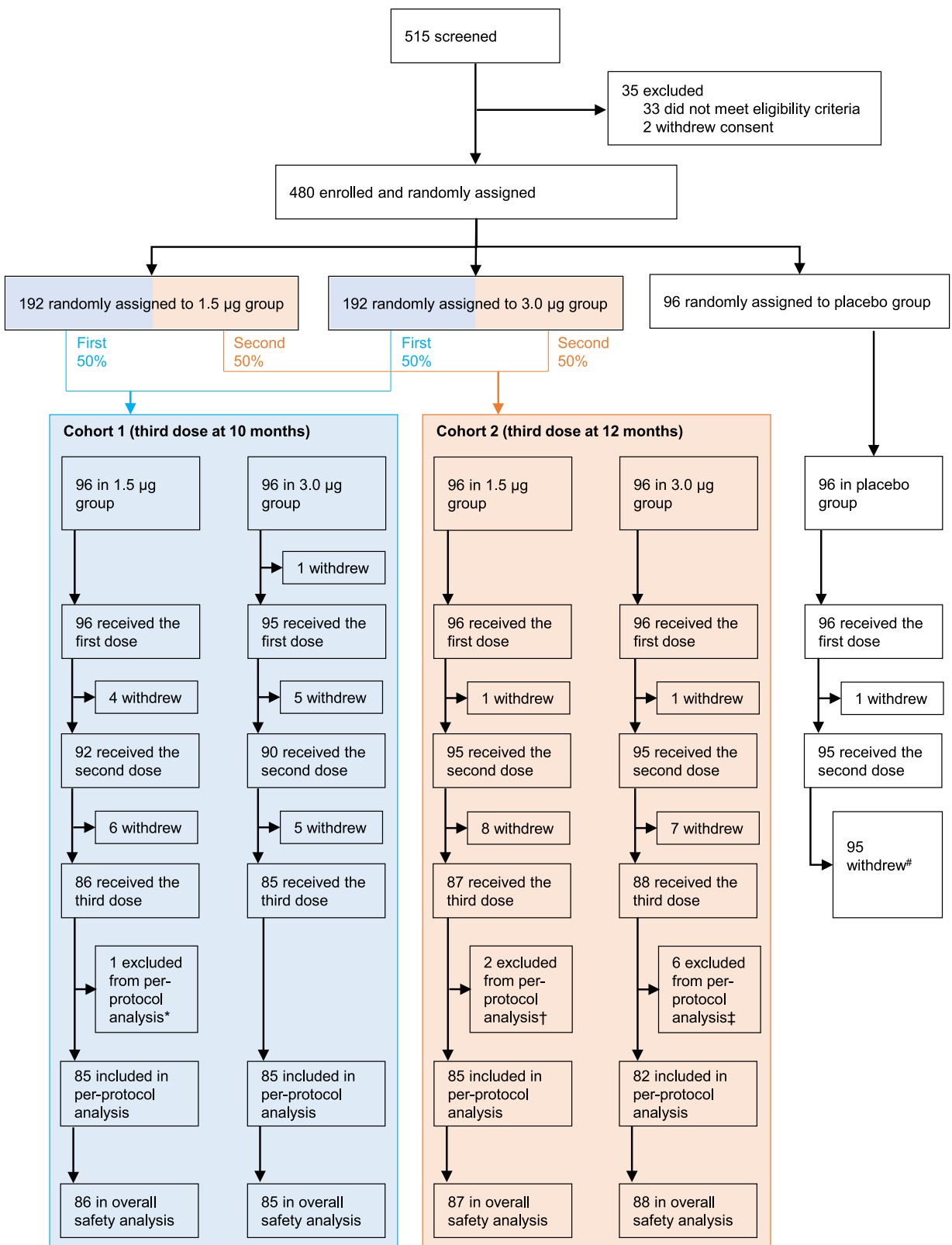

**Fig. 1 | Flow Chart.** # Participants in placebo group withdrew from the study and received COVID-19 vaccines since the inactivated COVID-19 vaccine was approved for emergency use among children and adolescents aged 3–17 years. *Participants in the 1.5 μg group were excluded from the per-protocol analysis because they did not have a blood sample taken 28 days after the third dose. †Participants in the 1.5 μg group were excluded from the per-protocol analysis because they received other licensed inactivated COVID-19 vaccine before the third dose, and one did not have a blood sample taken 28 days after the third dose. ‡Participants in the 3.0 μg group were excluded from the per-protocol analysis because they received other licensed inactivated COVID-19 vaccine before the third dose, and five did not have a blood sample taken 28 days after the third dose.

**Table 1 | Baseline demographic characteristics of participants who received the third dose**

| | Cohort 1 (n = 171) | | Cohort 2 (n = 175) | |
|---|---|---|---|---|
| | 1.5 µg group (n = 86) | 3.0 µg group (n = 85) | 1.5 µg group (n = 87) | 3.0 µg group (n = 88) |
| Age, years | 9.4 (3.7) | 9.0 (3.7) | 9.3 (3.8) | 9.4 (3.8) |
| 3–5 | 20 (23%) | 21 (25%) | 21 (24%) | 20 (23%) |
| 6–11 | 33 (38%) | 32 (38%) | 32 (37%) | 34 (39%) |
| 12–17 | 33 (38%) | 32 (38%) | 34 (39%) | 34 (39%) |
| Sex | | | | |
| Male | 46 (53%) | 42 (49%) | 44 (51%) | 52 (59%) |
| Female | 40 (47%) | 43 (51%) | 43 (49%) | 36 (41%) |

Data are n (%) or mean (SD).

Severities of solicited local and systemic adverse reactions reported within 28 days after the third dose were Grade 1 (mild) and Grade 2 (moderate) in both cohorts. Most adverse reactions occurred within 7 days after a booster dose. The most commonly reported reactions were injection-site pain (14 [4.1%] participants) and fever (5 [1.5%] participants). The overall incidences of adverse reactions within 28 days after the third dose were 14 (8.1%) of 173 participants in the 1.5 µg group and 15 (8.7%) of 173 participants in the 3.0 µg group (Table 3).

Only one participant in the 1.5 µg group had reported a serious adverse event (acute tonsillitis; Supplementary p 4), which was unrelated to vaccination. The incidence of serious adverse event was 0.3% (one of 346 participants) within 28 days after the third dose. From the beginning of enrolment to 28 days after the third dose in this trial, no adverse events of special interest occurred in this study. The follow-up of serious adverse events and adverse events of special interest is ongoing.

## Discussion

To our knowledge, this is the first report of the safety and immunogenicity in children and adolescents who received a third dose inactivated COVID-19 vaccine. We found that a third dose of CoronaVac was well-tolerated and induced anamnestic responses against SARS-CoV-2. Regarding the well-tolerated of the booster dose, the incidence of adverse reactions in the two dose groups was similar, indicating no

dose-related concern on safety. Moreover, the incidence of adverse reactions following the third dose was not higher than that of the first two doses. All reactions were transient, mild to moderate in severity. Injection-site pain was the most frequently reported symptom. Only one participant in the 1.5 µg group with a serious adverse event was a case of tonsillitis that was unrelated to vaccination. None of adverse events of special interest occurred in this trial as of now.

Our study showed that the initial neutralising antibody levels against the prototype induced by the first two doses decreased over time, which was also observed in our previous study in adults and the elderly[17]. However, the antibody levels were higher in children than that in adults and the elderly at the same time points. This is likely due to the antibody titre peak on day 28 after two-dose primary immunisation in children being higher than that in adults and elderly[13–15]. The waning immune humoral response was also found in Moderna's mRNA-1273 vaccine and Pfizer's BNT162b2 vaccine[19,20]. A homologous booster dose of CoronaVac led to a robust immune anamnestic response in children and adolescents aged 3–17 years. Immune responses induced by a 3.0 µg booster dose were higher than those induced by a 1.5 µg booster dose in cohort 2. Booster at 12 months after dose 2 induced a higher neutralising antibody titre than booster at 10 months within the 3.0 µg groups. The increase in neutralising antibody titres against prototype around 33-fold increases compared with pre-booster and around 5-fold increases compared with 28 days after dose 2. The seropositivity rate against the prototype was 100% regardless of the dose group after the third dose. This anamnestic immune response in children was much more higher levels of neutralising antibody response than previously reported in adults and the elderly. The results implied that age might play a vital role in the neutralising antibody response to the vaccine, and a lower dose vaccine could induce a higher immune memory response when compared with adults and the elderly.

Evidence from various studies supports the Omicron variant escapes humoral immunity induced by natural infection or vaccination[3,7,8,21,22]. However, a real-world study in Hong Kong had shown that three doses of vaccines (COMIRNATY or CoronaVac) offered a very high level of protection against severe outcomes (Hospitalisation or death) in the Omicron variant pandemic, which showed that a third dose vaccine could provide additional protection in adults[10]. Our study found that participants boosted with 3.0 µg CoronaVac at 12-month interval exhibited neutralisation of Omicron

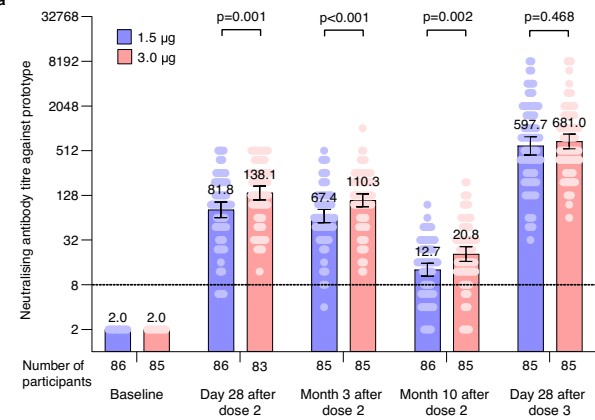

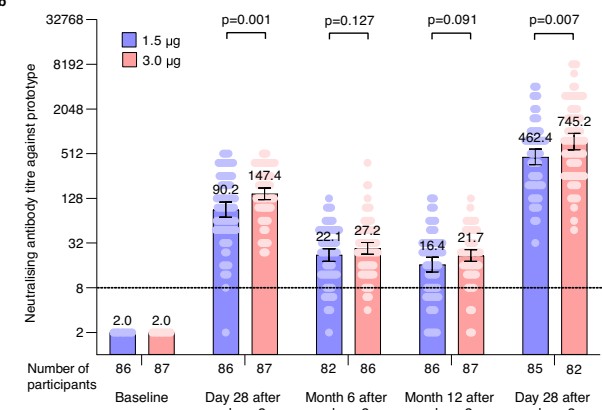

**Fig. 2 | Levels of neutralising antibody against the prototype SARS-COV-2 after immunogenicity assessment. a** Neutralising antibody titres against the prototype SARS-COV-2 in Cohort 1. **b** Neutralising antibody titres against the prototype SARS-COV-2 in Cohort 2. The number of participants for 1.5 µg group (blue) and 3.0 µg group (red) at each visit included in the analysis is provided below the bars. Dots are reciprocal neutralising antibody titres for individuals in the per-protocol population. Numbers above the bars are GMTs, and the error bars indicate the 95%CI.

GMTs and corresponding 95%CI were calculated on the basis of standard normal distributions of log-transformed antibody titres. Numbers above the short horizontal lines are p values of comparisons between 1.5 µg group and 3.0 µg group using group t test with log-transformation (two-sided). The dotted horizontal line represents the seropositivity threshold (1:8). Titres lower than the limit of detection (1:4) are presented as half the limit of detection. GMT geometric mean titre.

**Table 2 | Immunogenicity assessment on day 28 after the third dose**

|  | Cohort 1 | | | Cohort 2 | | |
|---|---|---|---|---|---|---|
|  | 1.5 µg group | 3.0 µg group | *p* value | 1.5 µg group | 3.0 µg group | *p* value |
| *Prototype strain* | | | | | | |
| Seropositivity n/N (%) (95%CI) | 85/85 (100.0) (95.8–100.0) | 85/85 (100.0) (95.8–100.0) | 1.000 | 85/85 (100.0) (95.8–100.0) | 82/82 (100.0) (95.6–100.0) | 1.000 |
| GMT (95%CI) | 597.7 (452.4–789.6) | 681.0 (545.2–850.7) | 0.468 | 462.4 (364.4–586.6) | 745.2 (577.0–962.3) | 0.007 |
| GMFR (95%CI) | 46.4 (35.3–60.9) | 32.8 (25.1–42.8) | 0.072 | 28.0 (22.0–35.6) | 33.3 (25.0–44.4) | 0.350 |
| *Omicron variant* | | | | | | |
| Seropositivity n/N (%) (95%CI) | 66/85 (77.7) (67.3–86.0) | 77/85 (90.6) (82.3–95.9) | 0.021 | 67/85 (78.8) (68.6–86.9) | 75/82 (91.5) (83.2–96.5) | 0.022 |
| GMT (95%CI) | 23.6 (17.7–31.4) | 34.5 (27.2–43.7) | 0.043 | 24.0 (18.0–32.2) | 40.0 (30.7–52.3) | 0.011 |
| GMFR (95%CI) | 11.5 (8.6–15.2) | 16.0 (12.8–20.0) | 0.066 | 11.2 (8.4–14.8) | 19.2 (14.8–24.8) | 0.006 |

*GMT* geometric mean titre, *GMFR* geometric mean fold rise, taking pre-booster as baseline.
Data are n/N (%) (95% CI) unless otherwise stated. Comparison between groups was conducted by group *t*-test with log-transformation. Pearson's chi-squared test and Fisher's exact test were used to analyse categorical outcome. Two-sided tests were used.

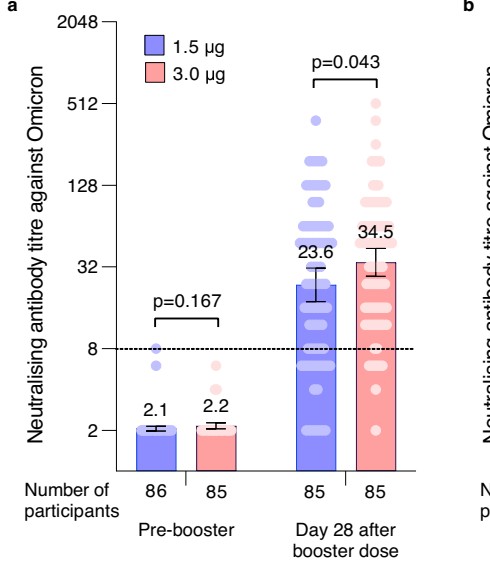

**Fig. 3 | Levels of neutralising antibody against the Omicron 28 days after the third dose. a** Neutralising antibody titres against the Omicron in Cohort 1. **b** Neutralising antibody titres against the Omicron in Cohort 2. The number of participants for 1.5 µg group (blue) and 3.0 µg group (red) at each visit included in the analysis is provided below the bars. Dots are reciprocal neutralising antibody titres for individuals in the per-protocol population. Numbers above the bars are GMTs, and the error bars indicate the 95% CI. GMTs and corresponding 95% CI were calculated on the basis of standard normal distributions of log-transformed antibody titres. Numbers above the short horizontal lines are p values of comparisons between 1.5 µg group and 3.0 µg group using group *t* test with log-transformation (two-sided). The dotted horizontal line represents the seropositivity threshold (1:8). Titres lower than the limit of detection (1:4) are presented as half the limit of detection. *GMT* geometric mean titre.

variant. The seropositivity rate against the Omicron was above 91%, although the GMTs were around 18-fold lower than against the prototype strain. These results suggested a third dose CoronaVac enhanced cross-reactivity of neutralising antibody responses. This finding was in line with the previous study where a third dose of inactivated COVID-19 vaccine induces cross-neutralising immunity against the SARS-CoV-2 Omicron variant[11,12,23]. Wang reported a third dose of inactivated vaccine augments the potency, breadth, and duration of anamnestic responses against SARS-CoV-2. Repeated antigen stimulation confers the development of monoclonal antibodies with enhanced neutralising and breadth[24].

In this study, we found a significant reduction (18.6- to 25.3-fold) of neutralising antibody titre against Omicron compared to the prototype on day 28 after the booster dose. The reduction of neutralising antibody titre against Omicron has been observed with other

COVID-19 vaccines. One report showed that individuals who received a booster dose of NVX-CoV2373, a SARS-CoV-2 recombinant spike protein vaccine, exhibited an average 15.9-fold reduction in neutralising antibody titres against BA.1 compared to ancestral virus when using a microneutralisation test[23]. Another study showed a reduction of 61.3-fold for Omicron strain at 21 days after two doses of BNT162b2 and a reduction of 6.35-fold at 1 month after three doses of BNT162b2[25]. Therefore, the Omicron variant escapes humoral immunity may be associated with a higher risk of breakthrough infection for COVID-19 vaccine recipients. Although evidence shows that protection of symptomatic Omicron infection from two doses current vaccines is significant, vaccine effectiveness against hospitalisation and severe disease may be well maintained after booster dose[10,26]. Booster with current vaccines increases the neutralisation potency better than that achieved with two-dose primary vaccination. It is probable that, even if

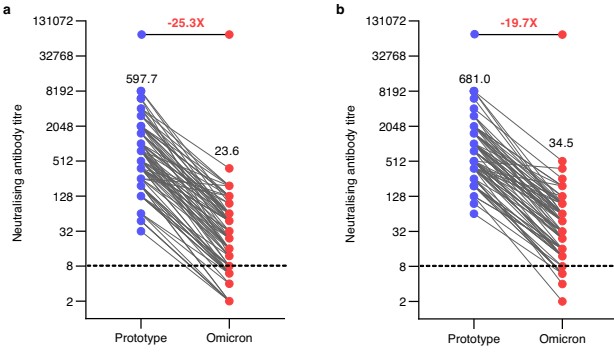
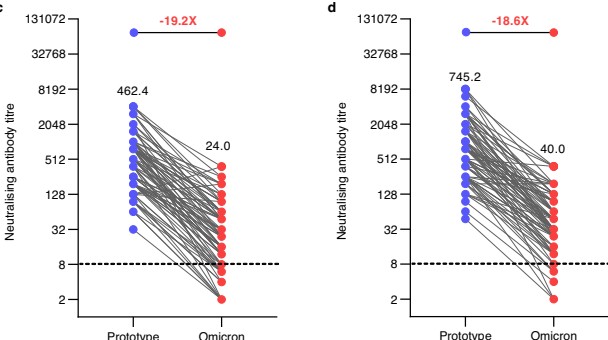

**Fig. 4 | Reduction of cross-neutralisation to the Omicron. a** Paired neutralising antibody titres against the prototype SARS-COV-2 (blue) and the Omicron (red) in Cohort 1, 1.5 µg group (*n* = 85). **b** Paired neutralising antibody titres against the prototype SARS-COV-2 (blue) and the Omicron (red) in Cohort 1, 3.0 µg group (*n* = 85). **c** Paired neutralising antibody titres against the prototype SARS-COV-2 (blue) and the Omicron (red) in Cohort 2, 1.5 µg group (*n* = 85). **d** Paired neutralising antibody titres against the prototype SARS-COV-2 (blue) and the Omicron (red) in Cohort 2, 3.0 µg group (*n* = 82). Dots are reciprocal neutralising antibody titres for individuals in the per-protocol population. Numbers above the dots are GMTs. Numbers above the short horizontal lines are reduction folders of cross-neutralisation comparisons between prototype and Omicron. The dotted horizontal line represents the seropositivity threshold (1:8). Titres lower than the limit of detection (1:4) are presented as half the limit of detection. *GMT* geometric mean titre.

pre-existing SARS-CoV-2 antibodies may poorly prevent Omicron infection, anamnestic responses and cellular immunity will be operative to prevent severe outcomes[27].

In an exploratory analysis stratified by age, we observe a significant difference in neutralising antibodies against Omicron among age groups after the third dose (Supplementary Table 3-2). GMTs in both cohorts decreased with increasing age in recipients of the same dose. This finding was similar to the GMTs against the prototype after two doses of CoronaVac. This finding is probably because children have a higher lymphocyte count, and lymphocytes play a pivotal role in adaptive immunity and are the core component of the immune system[28]. Moreover, the reduction folds of neutralising GMTs against live SARS-CoV-2 Omicron in both cohorts were minimal in the 3–5 years subgroup compared with other age subgroups (Supplementary Fig. 1). This observation may be in part

### Table 3 | Adverse reactions within 28 days after the third dose

| | 1.5 µg group (*n* = 173) | 3.0 µg group (*n* = 173) | Total (*n* = 346) |
|---|---|---|---|
| *Any adverse reaction* | | | |
| Total | 14 (8.1) | 15 (8.7) | 29 (8.4) |
| Grade 1 (mild) | 9 (5.2) | 11 (6.4) | 20 (5.8) |
| Grade 2 (moderate) | 5 (2.9) | 4 (2.3) | 9 (2.6) |
| *Systemic diseases and injection site adverse reactions* | | | |
| Injection site pain | 5 (2.9) | 9 (5.2) | 14 (4.1) |
| Fever | 3 (1.7) | 2 (1.2) | 5 (1.5) |
| Injection site itch | 3 (1.7) | 1 (0.6) | 4 (1.2) |
| Injection site swelling | 0 | 1 (0.6) | 1 (0.3) |
| Fatigue | 0 | 1 (0.6) | 1 (0.3) |
| *Respiratory, thoracic, and mediastinal disorders* | | | |
| Cough | 4 (2.3) | 1 (0.6) | 5 (1.5) |
| Runny nose | 0 | 1 (0.6) | 1 (0.3) |
| *Skin and subcutaneous tissue disorders* | | | |
| Itch | 1 (0.6) | 1 (0.6) | 2 (0.6) |
| Erythema | 1 (0.6) | 0 | 1 (0.3) |
| Rash | 0 | 1 (0.6) | 1 (0.3) |
| *Gastrointestinal disorders* | | | |
| Nausea | 1 (0.6) | 0 | 1 (0.3) |
| Abdominal pain | 0 | 1 (0.6) | 1 (0.3) |

Data are *n* (%), representing the total number of participants who had adverse reactions (i.e., adverse events related to vaccination).

because the cross-neutralising activity correlates with the magnitude of neutralising antibody responses.

Research data is urgently needed to optimize the interval between the booster and primary immunisation for children and adolescents in the Omicron context. A short interval may not induce a robust anamnestic response since the memory B cells were immature[29]. A more than 6 months interval induced a higher concentration of netralising antibodies in adults. Our study showed that booster with 3.0 ug dose of CoronaVac at 12 months induced higher netralising antibody titers against the Omicron than that at 10 months. Of note, the seropositivity rate against the Omicron was both above 90% in 3.0 µg group and against prototype were 100% in any dose group, either with a 10-month interval or with a 12-month interval between the booster and primary immunisation. This finding confirmed that a homologous booster dose can enhance the antibody cross-protection against Omicron.

There are several limitations to this study. First, we assessed immune response against the Omicron variant only focusing on neutralising antibody response. Other humoral responses, for instance, anti-spike IgG, anti-nucleocapsid, etc. were not assessed in our study. Cellular immune response, for instance, T-cell immunity, might contribute to protection[30,31]. The specific T cells induced by a booster dose of CoronaVac recognize Delta and Omicron variants of SARS-CoV-2 was observed in adults[11]. However, T-cell response was not assessed in our study, and this needs to be further studied in children. Second, although neutralising antibody titres were higher than in adults and the cross-neutralising response against Omicron also observed, protection against the Omicron variant and persistence protection in a booster with CoronaVac need to be continually evaluated in further real-world studies. Third, the sample size of this booster study is relatively small per age group. A larger, multicenter clinical study will be needed to evaluate to provide scientific evidence for booster strategy in children and adolescents. Finally, we report the results of interim analyses 28 days after the booster dose. Long-term follow-up is ongoing to identify the duration of immunity induced by the third dose and to assess the long-term safety.

In conclusion, a third dose of CoronaVac is safe and induces a robust immune anamnestic response to prototype SARS-CoV-2 in children and adolescents 10 months or 12 months after the second dose, resulting in a significant increase in the concentration of neutralising antibodies, and might have neutralising potency against the Omicron variant in children and adolescents.

## Methods

### Study design and participants

This study was carried out based on the double-blind, randomised, placebo-controlled, phase 2 clinical trial initiated on 12 December 2020, in healthy children and adolescents aged 3–17 years in Zanhuang County, Hebei Province. The design of this phase 2 clinical trial had been published previously[14]. Briefly, all eligible participants were assigned randomly to 1.5 μg, 3.0 μg, or placebo groups in a ratio of 2:2:1. Randomisation codes were generated by the randomisation statistician by means of block randomisation using SAS software version 9.4. The randomisation code was allocated to each participant in sequence in the order of enrolment, and then the participants received the investigational products labelled with the same code. Concealed random group allocations and blinding codes were kept in signed envelopes. Investigators, participants, and laboratory staff were masked to group allocation. To evaluate the immunogenicity of primary vaccination, blood samples were taken before vaccination and on day 28 after the second dose. To evaluate the antibody persistence following primary vaccination and the immune response of a booster dose in the different intervals, the protocol (Version 2.5) was adapted. For evaluation of cross-neutralising immunity with SARS-CoV-2 Omicron variant after a booster dose, the protocol (Version 2.6) was amended again on 18 February 2022. The amended protocols were updated on ClinicalTrials.gov.

According to the amended protocol, participants in placebo groups withdrew from the study and received vaccines since the inactivated COVID-19 vaccine was approved for emergency use in children and adolescents aged 3–17 years on 11 June 2021, in China[18]. According to their random number order, the first half of participants in the vaccine group were assigned to cohort 1, while the rest of the second half of participants in the vaccine group were assigned to cohort 2. In cohort 1, participants received a booster dose of vaccine 10 months after the second dose (with a 30-day window period). In cohort 2, participants received a booster dose of vaccine 12 months after the second dose (with a 30-day window period). The dosage of booster dose was the same as their first two doses. Key exclusion criteria for the third dose are also the same as the second dose. Participants with COVID-19 infections or given with other COVID-19 vaccines were excluded. Written informed consent was obtained from all participants' parents, and participants 8–17 years of age also provided written assents before inoculation of a booster dose in eligible participants. The amended protocols and informed consent form for the study were approved by Hebei CDC Ethics Committee (IRB2021-414, IRB2022-078).

To evaluate the antibody persistence following primary vaccination, blood samples were collected at 3 months and 10 months after the second dose in cohort 1. While for cohort 2, blood samples were collected at 6 months and 12 months after the second dose. For participants who received a booster dose in both cohorts, blood samples were collected 28 days after the booster dose to assess the immunogenicity to both the prototype and Omicron strains (Fig. 1).

Safety information after the third dose was obtained using the same methods as the first two doses. Briefly, for the first 7 days after the third dose, participants were required to record the injection-site adverse events (e.g. pain, pruritus, swelling, and redness), or systemic adverse events (e.g. fever, headache, cough, and allergic reaction) on the diary cards. On day 8 and day 28, participants visited the study site for assessment by the study investigators (all medical practitioners) who conducted face-to-face interviews to confirm safety. Between visits, safety data were collected by spontaneous recording and reporting of adverse events by participants. Participants use contact cards and phone to investigators to report their AEs. For the young children, their parents will observe, report their child's AEs, and fill in diary card or contact card. We also planned to collect serious adverse events (SAEs) and adverse events of special interest (AESIs) until

12 months after the third dose. Reported adverse events were graded according to the National Medical Products Administration Guidelines for grading standards of adverse events in clinical trials of preventive vaccines (2019)[32]. The grade includes Grade 1 (mild), Grade 2 (moderate), Grade 3 (severe), Grade 4 (critical), and Grade 5 (death).

Immunological assessment methods and related procedures were described in Supplementary information (p 2). Neutralising antibodies against infectious prototype SARS-CoV-2 (virus strain SARS-CoV-2/human/CHN/CN1/2020, GenBank accession number MT407649.1) and SARS-CoV-2 Omicron variant (CHK07) were quantified using a microcytopathogenic effect assay. Several measures were taken to control the quality of the microcytopathogenic effect assay, including virus back-titration for each batch of tests to determine whether the amount of virus was within the range of 32–320 tissue culture infectious dose ($TCID_{50}$) per 50 μL. Two types of positive antibody control, a negative antibody control, a serum toxicity control, and a cell control were included for each test. Detection was done by the National Institute for Food and Drug Control.

Electronic Data Capture (EDC) RIEHEN (Version: 2.1.1610) was used to establish the electronic Case Report Form (eCRF) to record clinical trial data. Information was inputted with standard language according to the EDC instructions and eCRF filling instructions.

### Outcomes

The primary endpoint of this study is safety and immunogenicity following a booster dose of CoronaVac. The secondary endpoint of this study is antibody persistence following the primary two-dose schedule. Exploratory immunological outcomes include seropositive rate and GMTs of neutralising antibody against the live prototype SARS-CoV-2 at 3 months, 6 months, 10 months and 12 months after the second dose. To explore the cross-reactivity of neutralising antibody responses, seropositive rate, GMTs, and GMFR of neutralising antibody against the Omicron strain of a third dose were measured. The positive cutoff of the titre for neutralising antibodies to live prototype SARS-CoV-2 and Omicron was 1:8. For immunogenicity analysis, we included the participants who received their assigned third doses and had available antibody results on day 28 after the third dose.

The safety endpoint was the incidence rate of adverse reactions that occurred within 28 days after the third dose. Secondary safety endpoints were SAEs and AESIs occurring from the first dose to 12 months after the third dose in both vaccination cohorts.

### Ethical statement

We complied with all relevant ethical rules. The complete study protocol was approved by the ethics committees of Hebei Provincial Centre for Disease Control and Prevention (IRB2020-005). The amended protocols and informed consent form for the booster dose study were approved by Hebei CDC Ethics Committee (IRB2021-414, IRB2022-078).

### Statistical analysis

We assessed immunological endpoints in the per-protocol population, which included all participants who completed their assigned third doses and had antibody results available according to the protocol. In addition, we assessed the antibody persistence following primary vaccination in the immune persistence analysis set, which included participants who completed 10-months follow-up after the second doses for cohort 1, and who completed 12-months follow-up after second doses for cohort 2. Safety assessments for the third dose were done in a safety population data set of all participants who received a third dose.

The demographics of participants who received the third dose were summarised for vaccination cohorts, and we used Pearson chi-squared test or Fisher's exact test to analyze categorical outcomes. We calculated 95% CIs for all categorical outcomes using the Clopper-

Pearson method. We figured GMTs and corresponding 95% CIs based on the standard normal distribution of log-transformed antibody titres. For the third dose given at 10 months or 12 months after the second dose, GMFR were calculated using antibody titres before booster vaccination and 28 days after the booster dose (taking pre-booster as baseline).

Comparisons were made between groups by group t-tests or ANOVA model with log-transformation (per GMT and GMFR). Hypothesis testing was two-sided, and we considered p values of less than 0.05 to be significant. We used SAS (Version 9.4, SAS Institute Inc., Cary, USA) for all analyses. The trials are registered with Clinical-Trials.gov, NCT04551547.

### Reporting summary

Further information on research design is available in the Nature Research Reporting Summary linked to this article.

## Data availability

The study protocol is available in the Supplementary Material. The individual participant-level data that underlie the results reported in this article will only be shared after de-identification (text, tables, figures, and supplementary). This clinical trial is ongoing, and all the individual participant data cannot be available until the immune persistence evaluation is conducted. Source data underlying all figures are provided with this paper. Researchers who provide a scientifically sound proposal will be allowed to access to the de-identified individual participant data. These proposals will be reviewed and approved by the sponsor, investigators and collaborators on the basis of scientific merit. To gain access, data requestors will need to sign a data access agreement. Proposals should be directed to gaoq@sino-vac.com. Source data are provided with this paper.

## Code availability

The SAS code for the main analysis is available on GitHub at https://github.com/wanglei365/sinovac_1003_antibody.

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

## Acknowledgements

This study was funded by the National Key Research and Development Program (2020YFC0849600) and Beijing Science and Technology Program (Z201100005420023). The funders had no role in study design, data collection and analysis, decision to publish, or preparation of the manuscript.

## Author contributions

Q.G., Y.Z., Le.W., and Z.W. designed the trial and study protocol. Le.W., Z.W., and M.L. contributed to the literature search. All authors had access to data, and H.Z., Li.W., and J.L. verified the data. Le.W. and Y.H. wrote the first draft manuscript. Q.G., Y.Z., Z.W., Y.H., and Le.W. contributed to the data interpretation and revision of the manuscript. Q.S. and Le.W. contributed to data analysis. Y.H. and Le.W. monitored the trial. H.W. and W.J. were responsible for the site work including the recruitment, follow-up and data collection and Z.W. was the site coordinator. Z.Y. were responsible for the laboratory analysis. All authors had final responsibility for the decision to submit the manuscript for publication.

## Competing interests

The authors declare no competing interests.
