## [Peer review file · Nature Communications]

REVIEWER COMMENTS

Reviewer #1 (Remarks to the Author):

This is an important clinical trial that presents both safety and neutralising antibody immunogenicity results against the Omicron SARS-CoV-2 variant of concern after a third dose of CoronaVac vaccine in children and adolescents, and looks at neutralising antibody persistence against a live prototype of SARS-CoV-2 after a primary dosing series. The data and analysis are of good quality, and are presented in a well written manuscript, providing important results to add to the currently limited body of evidence on COVID-19 vaccines and booster doses in children and adolescents.

I do have some comments, and suggestions to increase the clarity for certain parts of text:

General comments:

1. Please go into further detail to describe the choice/reasoning behind the intervals. Why are they 10 and 12 months – what was the reasoning behind these very similar intervals? There could be a lot of overlap given the 30-day windows, and there is no real short/long distinction, for example.
2. The study has shown good persistence of neutralising antibodies after two doses of CoronaVac, which was not seen in previous reports in an older population. E.g. [https://doi.org/10.1016/S0140-6736\(22\)00094-0](https://doi.org/10.1016/S0140-6736(22)00094-0). It would be interesting to add these comparisons into the discussion. How do the assay readings in this paper compare to those in other papers?
3. Please comment on the results of the cross-neutralising activity to that seen in other papers. Are the results consistent with previous studies?
4. The lack of T-cell data is addressed in the limitations; the lack of data on other humoral responses, e.g. anti-spike IgG, anti-nucleocapsid, etc is also a limitation of this study. In the discussion there are a lot of statements on humoral responses. As the study only presented neutralising antibody results, it would be appropriate to be specific to avoid overinterpretation of the results.

Could other assays be run with the immunological blood samples to explore humoral immunogenicity results beyond just neutralising antibody? (Whilst not answering the question regarding immune response against the Omicron variant, it would be interesting to see the waning of other humoral responses following the primary immunisations, and compare with other studies.)
5. Please quote more of the 95% CIs in the text rather than just effect sizes.
6. The study is unlikely to be sufficiently powered to detect most safety events reported in Table 3. I strongly suggest to remove the p-values reported here. It is sufficient to descriptively present these outcomes in the manuscript with proportions, as has mostly been done.

7. The supplementary page references in the text need updating as they currently don't point to the correct page (for example on line 97).

8. The allocation of participants to the 2 cohorts could be better described (line 251-254). In Figure 1 it looks like the order of initial randomisation dictates which cohort a participant is placed into; however, in the text it just states that half the participants are allocated to cohort 1 and half to cohort 2. Please expand on this in the text so that it is clear and matches the information in the figure.

Specific comments:

Abstract

9. On lines 22 and 32-33, I would suggest to clarify what is meant by primary and booster vaccination e.g. "booster (third dose) and two-dose primary immunisation" (line 22) and "two-dose primary and booster vaccination" (line 32-33).

10. In the abstract the third dose is often referred to as a "booster" dose (lines 25, 27, 34) but later as "dose 3". Be consistent with the terminology, perhaps first stating "3rd dose booster" to be clear and then "booster" for all subsequent mentions.

11. Please remove the subjective words "remarkably" and "remarkable" (lines 34 and 37), replacing with "significantly" where results are statistically significant.

12. Please add 95% confidence intervals to point estimates (lines 34 and 35).

Introduction

13. Clarify that this is discussing a two-dose primary vaccination and specify that the booster dose is dose 3 (line 47).

14. Line 52 – in which country/ies is this the schedule? Please clarify.

Methods

15. Line 250-251 – wording unclear. Suggest changing to "... was approved for emergency use in children and adolescents aged 3-17 years..."

16. Line 255 – it is stated that key exclusion criteria is the same but it would be useful to readers to explicitly state in the text that subjects with COVID-19 infections were excluded.

17. How the consent was obtained for children under 8 years?

18. Line 269 – was this spontaneous reporting still diary-based, or left until a follow-up visit to report? Please clarify. And how did the young children report their AEs? Would be useful to summarise this here.

19. Line 271-273 – suggest to add the scale on which they were graded e.g. mild (grade 1), moderate (grade 2), severe (grade 3). The results discuss “mild and moderate” reactions so it would be useful to add this into the methods section.

20. Formal hypothesis testing has been performed for the occurrence of solicited adverse events (table 3) but this isn't clearly stated in the statistical analysis section. Lines 310-312 do cover this but need to more explicitly state that this includes the solicited AE outcomes, for clarity. However, this point is irrelevant if the suggestion in point 6 is followed (formal hypothesis testing not advised for this outcome).

Results

21. Line 63 – for clarity, suggest stating randomly allocated.

22. Line 66 – suggest changing to “... was approved for emergency use among children and adolescents aged 3-17 years.”

23. Line 73 – either median or IQR is incorrect: median of 360 days is outside of IQR 362-363 days.

24. Line 83-84 – no natural infections reported from the beginning of the trial. Is this the entire trial (i.e. from the primary vaccinations) or this sub-study of just the third dose? Suggest to clarify.

25. Line 86 – suggest to spell out geometric mean titre (GMT) here as it's the first time mentioning the abbreviation in the main body text.

26. Line 87 & 88 – add 95% CIs for the seropositivity rates.

27. Line 91 – please change the subjective word “remarkably” to “significantly”.

28. Line 95 – add 95% CIs for the fold-increases.

29. Line 95-97 – this wording is unclear. “...in the 3.0µg group at day 28 after the third dose compared with that on day 28 after the second dose...” – needs to be clearer which timepoints are being compared.

30. Line 97 – suggest to use geometric mean fold rise (GMFR) instead of GMI. Increase can be absolute increase.

31. Line 102 – the text states that neutralising antibody titres declined to 16.4 at 12 months after dose 2 but in the figure it's 16.8.

32. Line 117 – please add 95% CIs when quoting the 11 and 16-fold increases.

33. Paragraph lines 122-137 – please state accompanying 95% CIs when quoting GMTs.

34. Line 139 – for clarity and to match the table, specify that mild is grade 1 and moderate is grade 2 (or do this in the table instead).

Discussion

35. Line 154 – hard to conclude “safe”, particularly when follow-up is still ongoing. Instead, I’d suggest stating that the third dose was well-tolerated and no safety concerns were raised.
36. Line 166-167 – consider using “higher” instead of “better”, or something else less subjective.
37. Line 168 – this is true only within the 3.0µg groups: GMT in cohort 1 1.5µg is higher vs GMT cohort 2 1.5µg when looking at prototype results. Need to explicitly state which group this is relating to (which then follows into line 169-170).
38. Line 172 – “robust” isn’t clear here when comparing with other studies. Please be more specific in the message here, e.g. higher levels of neutralising antibody response.
39. Line 174-175 – “and a longer interval and lower dose vaccine could induce higher immune memory response than adults and the elderly” – Don’t agree that the data supports this statement. In particular, the interval timings are not very different.
40. Line 178 – which vaccines? Please provide the names of these vaccines in brackets.
41. Line 181 – need to explicitly state here that this is for the 3.0µg dose, as the rate is around 78% when boosting with the 1.5µg dose.
42. Line 199-200 – Is there any reference to support this explanation?
43. Line 209 – typo of “moths” instead of “months”.

Figures and tables

44. Figure 1 – for clarity, I suggest adding the timing of third dose in brackets after Cohort 1 and Cohort 2, or in the “XX received the third dose” boxes. (E.g. either “Cohort 1 (third dose at 10 months)” or “XX received the third dose at the 10-month timepoint”.)
- Use “they” instead of “he” for the exclusions.
45. Figure 2 – be more specific in the title like in the other figures, i.e. after “immunogenicity assessment” state that this is neutralising antibody response.
 46. Table 3 – add in footnote the test used for p-values (if point 6 isn’t followed and the p-values are kept in the table).
 47. Serious adverse events on page 4 of supplementary file – the SAE is for a female but a lot of the pronouns state “he”. Please change to either “she” or “they”.

Reviewer #2 reviewed your manuscript in collaboration with reviewer #1.

Reviewer #3 (Remarks to the Author):

The authors have designed the pediatric trial with clear objectives. The results presented are convincing that the 3rd dose of vaccine elicits a significant increase in functional responses to Coronavac. Methodology used to measure immunogenicity is unique for the vaccine platform.

There are some gaps noticed in the data presented by the authors.

With some additional data analysis and clarification in the text this work can be accepted for publication.

1. Relevance of the prototype vaccine responses and protection against current Omicron variant (BA.5) needs to be addressed.

2. There is a striking difference in response with the younger children generating more antibodies compared to older adolescents.

But GMT presented in Figure 2 and 3 does not reflect this. A break up of the individual age group contribution in a bar diagram will help understand this better than the table. Does this apply for prototype and the variants needs to be addressed.

3. Since this pandemic has been a moving target do the authors think booster dose after 10 or 12 months is relevant; the immune responses from the prototype is not preventing breakthroughs and infection but only serious illness and hospitalization. Please add this in the discussion.

REVIEWER COMMENTS

Reviewer #1 (Remarks to the Author):

This is an important clinical trial that presents both safety and neutralising antibody immunogenicity results against the Omicron SARS-CoV-2 variant of concern after a third dose of CoronaVac vaccine in children and adolescents, and looks at neutralising antibody persistence against a live prototype of SARS-CoV-2 after a primary dosing series. The data and analysis are of good quality, and are presented in a well written manuscript, providing important results to add to the currently limited body of evidence on COVID-19 vaccines and booster doses in children and adolescents.

I do have some comments, and suggestions to increase the clarity for certain parts of text:

We thank the reviewer for the positive assessment of our manuscript and for her/his many useful comments that helped improve the manuscript.

General comments:

1. Please go into further detail to describe the choice/reasoning behind the intervals. Why are they 10 and 12 months – what was the reasoning behind these very similar intervals? There could be a lot of overlap given the 30-day windows, and there is no real short/long distinction, for example.

We apologize for the lack of clarity for the assay detail. In our previous study, we found that antibody levels in children were higher than those in adults and the elderly at 6-8 months after a primary schedule. Due to the waning antibody levels over time, whether giving a booster with 10-12 months interval is reasonable in children should be evaluated. In our result there is no overlap on booster time point in cohort 1 (range 300-329 days) and cohort 2 (range 361-390 days).

2. The study has shown good persistence of neutralising antibodies after two doses of CoronaVac, which was not seen in previous reports in an older population. E.g. [https://doi.org/10.1016/S0140-6736\(22\)00094-0](https://doi.org/10.1016/S0140-6736(22)00094-0). It would be interesting to add these comparisons into the discussion. How do the assay readings in this paper compare to those in other papers?

Thank you for pointing this out to us. It is an interesting finding in our study. The antibody levels were higher in children than in adults and the elderly at the same time points (for instance on day 28 or 6 months after two-dose primary vaccination). That's why we chose a longer interval (for instance 10 or 12 months) booster schedule in children. In our discussion section, we revised the statement and gave an explanation that *this is likely due to the antibody titre peak on day 28 after two-dose primary immunisation in children is higher than that in adults and elderly.*

3. Please comment on the results of the cross-neutralising activity to that seen in other papers. Are the results consistent with previous studies?

We apologize for the lack of clarity. We comment on the results of the cross-neutralising activity in discussion section. *This finding was in line with the previous study where a third dose of inactivated COVID-19 vaccine induces cross-neutralising immunity against the SARS CoV-2 Omicron variant. One report showed that individuals who received a booster dose of NVX-CoV2373, a SARS-CoV-2 recombinant spike protein vaccine, exhibited an average 15.9-fold reduction in neutralising antibody titres against BA.1 compared to ancestral virus when using a microneutralisation test.*

References are:

1.Schultz BM, Melo-González F, Duarte LF, et al. A Booster Dose of CoronaVac Increases

Neutralizing Antibodies and T Cells that Recognize Delta and Omicron Variants of Concern. *mBio*. 13, (2022).

2. Yu X, Qi X, Cao Y, et al. Three doses of an inactivation-based COVID-19 vaccine induces cross-neutralizing immunity against the SARS CoV-2 Omicron variant. *Emerging Microbes & Infections* 11, (2022).

3. Mallory RM, Formica N, Pfeiffer S, et al. Safety and immunogenicity following a homologous booster dose of a SARS-CoV-2 recombinant spike protein vaccine (NVX-CoV2373): a secondary analysis of a randomised, placebo-controlled, phase 2 trial. *Lancet Infect Dis*. 22, (2022).

4. The lack of T-cell data is addressed in the limitations; the lack of data on other humoral responses, e.g. anti-spike IgG, anti-nucleocapsid, etc is also a limitation of this study. In the discussion there are a lot of statements on humoral responses. As the study only presented neutralising antibody results, it would be appropriate to be specific to avoid overinterpretation of the results.

Could other assays be run with the immunological blood samples to explore humoral immunogenicity results beyond just neutralising antibody? (Whilst not answering the question regarding immune response against the Omicron variant, it would be interesting to see the waning of other humoral responses following the primary immunisations, and compare with other studies.) We apologize for the lack of clarity and thank you for pointing this out to us. The lack of data on other humoral responses e.g. anti-spike IgG, anti-nucleocapsid, etc is really a limitation of this study. We revised our statements on humoral responses in our manuscript to avoid overinterpretation. For example, change “humoral responses” into “*neutralising antibody response*”. Since this study is the interim results of phase II clinical trials, the protocol determined the main outcome. we only tested the neutralising antibody. Maybe we can explore other humoral responses in the future study.

5. Please quote more of the 95% CIs in the text rather than just effect sizes.

Thank you for pointing this out to us. We revised accordingly in the text.

6. The study is unlikely to be sufficiently powered to detect most safety events reported in Table 3. I strongly suggest to remove the p-values reported here. It is sufficient to descriptively present these outcomes in the manuscript with proportions, as has mostly been done.

Thank you for pointing this out to us. We accept this suggestion. Just descriptively present these outcomes and removed the p value in table 3.

7. The supplementary page references in the text need updating as they currently don't point to the correct page (for example on line 97).

Thanks very much for pointing out a mistake in the text. We revised the page in main text.

8. The allocation of participants to the 2 cohorts could be better described (line 251-254). In Figure 1 it looks like the order of initial randomisation dictates which cohort a participant is placed into; however, in the text it just states that half the participants are allocated to cohort 1 and half to cohort 2. Please expand on this in the text so that it is clear and matches the information in the figure.

We apologize for the lack of clarity and thank you for pointing this out to us. We added a description for this. *According to their random number order, the first half of participants in the vaccine group were assigned to cohort 1, while the rest of the second half of participants in the vaccine*

group were assigned to cohort 2.

Specific comments:

Abstract

9. On lines 22 and 32-33, I would suggest to clarify what is meant by primary and booster vaccination e.g. “booster (third dose) and two-dose primary immunisation” (line 22) and “two-dose primary and booster vaccination” (line 32-33).

Thank you for your suggestion, we revised accordingly.

10. In the abstract the third dose is often referred to as a “booster” dose (lines 25, 27, 34) but later as “dose 3”. Be consistent with the terminology, perhaps first stating “3rd dose booster” to be clear and then “booster” for all subsequent mentions.

Thank you for your suggestion, we revised accordingly.

11. Please remove the subjective words “remarkably” and “remarkable” (lines 34 and 37), replacing with “significantly” where results are statistically significant.

Thanks very much for your comment. We have changed “remarkably” into “significantly”.

12. Please add 95% confidence intervals to point estimates (lines 34 and 35).

Thank you for pointing this out to us. The 95% confidence intervals to point estimates is 40.0 (95% CI 30.7-52.3). We revised abstract per formatting instructions, so this statement was removed.

Introduction

13. Clarify that this is discussing a two-dose primary vaccination and specify that the booster dose is dose 3 (line 47).

We apologize for the lack of clarity and thank you for pointing this out to us. We revised accordingly.

14. Line 52 – in which country/ies is this the schedule? Please clarify.

Thank you for pointing this out to us. We think this part we can also clarify the vaccines since the schedule is based on products. These schedules are 3-8 weeks interval for inactivated COVID-19 vaccine (CoronaVac and BBIBP-CorV) in China and 3 weeks apart for mRNA COVID-19 vaccine (COMIRNATY) in the USA.

Methods

15. Line 250-251 – wording unclear. Suggest changing to “... was approved for emergency use in children and adolescents aged 3-17 years...”

Thank you for your suggestion, we revised accordingly.

16. Line 255 – it is stated that key exclusion criteria is the same but it would be useful to readers to explicitly state in the text that subjects with COVID-19 infections were excluded.

Thanks very much for your comment. It is important to state this key exclusion criteria although there is no COVID-19 infections in our study. We have added it in this section.

17. How the consent was obtained for children under 8 years?

According to the GCP of China, written informed consent was obtained from all participants' parents. For children under 8 years, we obtained their oral consent according to the extent of their ability to understand. When their ages reach 8 years, we obtained new written consent.

18. Line 269 – was this spontaneous reporting still diary-based, or left until a follow-up visit to report? Please clarify. And how did the young children report their AEs? Would be useful to summarise this here.

We apologize for the lack of clarity. It is still diary-based report using contact card. Participants use contact cards and phone to investigators to report their AEs. For the young children, their parents will observe, report their child's AEs, and fill in diary card or contact card.. We clarify this part in the revised manuscript.

19. Line 271-273 – suggest to add the scale on which they were graded e.g. mild (grade 1), moderate (grade 2), severe (grade 3). The results discuss “mild and moderate” reactions so it would be useful to add this into the methods section.

Thank you for your suggestion, we have added the scale on the grade in this methods section. The grade includes Grade 1 (mild), Grade 2 (moderate), Grade 3 (severe), Grade 4 (critical), and Grade 5 (death).

20. Formal hypothesis testing has been performed for the occurrence of solicited adverse events (table 3) but this isn't clearly stated in the statistical analysis section. Lines 310-312 do cover this but need to more explicitly state that this includes the solicited AE outcomes, for clarity. However, this point is irrelevant if the suggestion in point 6 is followed (formal hypothesis testing not advised for this outcome).

Thanks very much for your comment. We accept suggestion in point 6. Just descriptively present these outcomes. We also revised the Table 3 (removed the p value).

Results

21. Line 63 – for clarity, suggest stating randomly allocated.

We revised accordingly.

22. Line 66 – suggest changing to “... was approved for emergency use among children and adolescents aged 3-17 years.”

We revised accordingly.

23. Line 73 – either median or IQR is incorrect: median of 360 days is outside of IQR 362-363 days. Thanks very much for pointing out a mistake in the text. We checked the raw data. this is a clerical error of median. It should be 363. The IQR is correct. We have revised the median in the manuscript.

24. Line 83-84 – no natural infections reported from the beginning of the trial. Is this the entire trial (i.e. from the primary vaccinations) or this sub-study of just the third dose? Suggest to clarify.

We apologize for the lack of clarity. There were no COVID-19 infections reported in this study. We changed “From beginning of this trial” into “From the primary vaccinations”.

25. Line 86 – suggest to spell out geometric mean titre (GMT) here as it's the first time mentioning the abbreviation in the main body text.

We revised accordingly.

26. Line 87 & 88 – add 95% CIs for the seropositivity rates.

We revised accordingly.

27. Line 91 – please change the subjective word “remarkably” to “significantly”.

We revised accordingly.

28. Line 95 – add 95% CIs for the fold-increases.

We revised accordingly.

29. Line 95-97 – this wording is unclear. “...in the 3.0µg group at day 28 after the third dose compared with that on day 28 after the second dose...” – needs to be clearer which timepoints are being compared.

We apologize for the lack of clarity. For clarifying the timepoints, we added the timepoints “Day 56” and “Day 356” into the sentence. Neutralising antibody concentrations 28 after booster (Day 356) were approximately 7.3-fold (95% CI 5.3-10.0) higher than neutralising antibody concentrations on day 28 after the second dose (Day 56) in the 1.5 µg group and approximately 4.9-fold (95% CI 3.7-6.4) in the 3.0 µg group (Supplementary p 5 table 3-1).

30. Line 97 – suggest to use geometric mean fold rise (GMFR) instead of GMI. Increase can be absolute increase.

Thank you for your suggestion. We have changed “GMI” into “GMFR” in the full text of manuscript including the tables.

31. Line 102 – the text states that neutralising antibody titres declined to 16.4 at 12 months after dose 2 but in the figure it's 16.8.

Thanks very much for pointing out a mistake in the text. We checked the raw data. this is a clerical error in figure. 16.4 is correct. We revised the figure.

32. Line 117 – please add 95% CIs when quoting the 11 and 16-fold increases.

For clarifying the 95% CIs, we changed the sentence into “The GMFR from before to after the booster dose were 11.5 (95% CI 8.6-15.2) in the 1.5 µg group and 16.0 (95% CI 12.8-20.0) in the 3.0 µg group (Fig. 3, table 2)”.

33. Paragraph lines 122-137 – please state accompanying 95% CIs when quoting GMTs.

We revised accordingly.

34. Line 139 – for clarity and to match the table, specify that mild is grade 1 and moderate is grade 2 (or do this in the table instead).

Thanks for your suggestion. We added the “mild” and “moderate” in the table 3. And this grade also added in methods section per point 19.

Discussion

35. Line 154 – hard to conclude “safe”, particularly when follow-up is still ongoing. Instead, I’d suggest stating that that the third dose was well-tolerated and no safety concerns were raised.

Thanks for your suggestion. We revised this statement.

36. Line 166-167 – consider using “higher” instead of “better”, or something else less subjective.

Thanks for your suggestion. We changed “better” into “higher”.

37. Line 168 – this is true only within the 3.0µg groups: GMT in cohort 1 1.5µg is higher vs GMT cohort 2 1.5µg when looking at prototype results. Need to explicitly state which group this is relating to (which then follows into line 169-170).

Thank you for pointing this out to us. It is only within the 3.0µg groups. We revised it and the follow sentence.

38. Line 172 – “robust” isn’t clear here when comparing with other studies. Please be more specific in the message here, e.g. higher levels of neutralising antibody response.

Thank you for your correction. In our study, we only tested neutralizing antibodies. We revised it as you suggestion.

39. Line 174-175 – “and a longer interval and lower dose vaccine could induce higher immune memory response than adults and the elderly” – Don’t agree that the data supports this statement. In particular, the interval timings are not very different.

Thank you for your correction. We revised the sentence by deleting the “longer interval”. In this sentence, we just stress the difference between children and adults or elder.

40. Line 178 – which vaccines? Please provide the names of these vaccines in brackets.

It is COMIRNATY or CoronaVac. We added the names of these vaccines in brackets.

41. Line 181 – need to explicitly state here that this is for the 3.0µg dose, as the rate is around 78% when boosting with the 1.5µg dose.

Thank you for your correction. We revised this sentence. Explicitly state this is for booster with the 3.0µg dose vaccine at 12-months interval.

42. Line 199-200 – Is there any reference to support this explanation?

Yes, we added the reference here and added a citation.

Reference is:

1.Zhang J, Xing S, Liang D, et al. Differential Antibody Response to Inactivated COVID-19 Vaccines in Healthy Subjects. Front Cell Infect Microbiol 2021.

43. Line 209 – typo of “moths” instead of “months”.

We revised accordingly.

Figures and tables

44. Figure 1 – for clarity, I suggest adding the timing of third dose in brackets after Cohort 1 and Cohort 2, or in the “XX received the third dose” boxes. (E.g. either “Cohort 1 (third dose at 10 months)” or “XX received the third dose at the 10-month timepoint”.)

Use “they” instead of “he” for the exclusions.

Thanks for your suggestion. We have revised it.

45. Figure 2 – be more specific in the title like in the other figures, i.e. after “immunogenicity assessment” state that this is neutralising antibody response.

Thanks for your suggestion. We have revised it.

46. Table 3 – add in footnote the test used for p-values (if point 6 isn’t followed and the p-values are kept in the table).

Thanks for your suggestion. We accept suggestion in point 6. Just descriptively present these outcomes. We also revised the Table 3 (removed the p value).

47. Serious adverse events on page 4 of supplementary file – the SAE is for a female but a lot of the pronouns state “he”. Please change to either “she” or “they”.

We revised accordingly.

Reviewer #2 reviewed your manuscript in collaboration with reviewer #1.

Thank you for your comments.

Reviewer #3 (Remarks to the Author):

The authors have designed the pediatric trial with clear objectives. The results presented are convincing that the 3rd dose of vaccine elicits a significant increase in functional responses to Coronavac. Methodology used to measure immunogenicity is unique for the vaccine platform.

There are some gaps noticed in the data presented by the authors.

With some additional data analysis and clarification in the text this work can be accepted for publication.

We thank the reviewer for the positive assessment of our manuscript and for her/his many useful comments that helped improve the manuscript.

1. Relevance of the prototype vaccine responses and protection against current Omicron variant (BA.5) needs to be addressed.

Thanks very much for your suggestions. To our knowledge, there is currently fewer evidence about prototype vaccine responses against Omicron subvariants **BA.5**. However, studies reported two-dose primary vaccination induced limited neutralisation of Omicron subvariants BA.1 and BA.2. The decrease in antibody efficacy helps to explain the high number of breakthrough infections of Omicron subvariants. However, a booster dose of vaccine, as well as vaccination of previously infected individuals, strongly increased overall levels of anti-SARS-CoV-2 neutralizing antibodies. It is probable that, even if pre-existing SARS-CoV-2 antibodies may poorly prevent Omicron infection, anamnestic responses and cellular immunity will be operative to prevent severe outcomes. We add statement in our manuscript as follows:

Although evidence shows that protection of symptomatic Omicron infection from two doses current vaccines is significant, vaccine effectiveness against hospitalisation and severe disease may be well maintained after booster dose. Booster with current vaccines increases the neutralisation potency better than that achieved with two-dose primary vaccination. It is probable that, even if pre-existing SARS-CoV-2 antibodies may poorly prevent Omicron infection, anamnestic responses and cellular immunity will be operative to prevent severe outcomes.

References are:

1. McMenamin ME, Nealon J, Lin Y, Wong JY. Vaccine effectiveness of two and three doses of BNT162b2 and CoronaVac against COVID-19 in Hong Kong. Preprint. medRxiv
2. Cheng SM, Mok CKP, Chan KC, et al. SARS-CoV-2 Omicron variant BA.2 neutralisation in sera of people with Comirnaty or CoronaVac vaccination, infection or breakthrough infection, Hong Kong, 2020 to 2022. Euro Surveill 2022.
3. Gagne M, Corbett KS, Flynn BJ, et al. Protection from SARS-CoV-2 Delta one year after mRNA-1273 vaccination in rhesus macaques coincides with anamnestic antibody response in the lung. Cell 2022.

2. There is a striking difference in response with the younger children generating more antibodies compared to older adolescents.

But GMT presented in Figure 2 and 3 does not reflect this. A break up of the individual age group contribution in a bar diagram will help understand this better than the table. Does this apply for prototype and the variants needs to be addressed.

Thanks very much for your suggestions. We did this exploratory analysis stratified by age in our supplementary figure 1. This figure maybe clearly showed the significant reduction of neutralising antibody titre against Omicron compared to the prototype within the individual age group.

3. Since this pandemic has been a moving target do the authors think booster dose after 10 or 12 months is relevant; the immune responses from the prototype is not preventing breakthroughs and infection but only serious illness and hospitalization. Please add this in the discussion.

Thanks very much for your suggestions. We added this in the discussion section. *Although evidence shows that protection of symptomatic Omicron infection from two doses current vaccines is significant, vaccine effectiveness against hospitalisation and severe disease may be well maintained after booster dose. Booster with current vaccines increases the neutralisation potency better than that achieved with two-dose primary vaccination. It is probable that, even if pre-existing SARS-CoV-2 antibodies may poorly prevent Omicron infection, anamnestic responses and cellular immunity will be operative to prevent severe outcomes.*

REVIEWERS' COMMENTS

Reviewer #1 (Remarks to the Author):

The authors have put a great deal of work into this well written manuscript, and we thank them for taking on board our previous comments and suggestions. We only have one further suggestion, which is to add to the text the booster ranges for each cohort stated in the authors' response to our first comment, so that readers can see there is no overlap in booster time points between the cohorts.

REVIEWER COMMENTS

Reviewer #1 (Remarks to the Author):

The authors have put a great deal of work into this well written manuscript, and we thank them for taking on board our previous comments and suggestions. We only have one further suggestion, which is to **add to the text the booster ranges for each cohort** stated in the authors' response to our first comment, so that readers can see there is no overlap in booster time points between the cohorts.

Thank you for your positive assessment of our manuscript. We really appreciate you taking time and effort to review our manuscript.

For clearer, we changed "IQR" into "range" in the main text.